# Unified View of Grokking, Double Descent and Emergent Abilities: A Comprehensive Study on Algorithm Task

**Yufei Huang**[1,2,3], **Shending Hu**[1,2,3], **Xu Han**[1,2,3], **Zhiyuan Liu**[1,2,3], **Maosong Sun**[1,2,3*]

[1]Department of Computer Science and Technology, Tsinghua University, Beijing
[2]Beijing National Research Center For Information Science And Technology
[3]Institute for Artificial Intelligence, Tsinghua University, Beijing
`huangyf530@gmail.com`

## Abstract

Recent studies have uncovered intriguing phenomena in deep learning, such as *grokking*, *double descent*, and *emergent abilities* in large language models, which challenge human intuition and are crucial for a deeper understanding of neural models. In this paper, we present a comprehensive study on algorithm task to provide a unified view of these three phenomena, with a focus on the interplay between memorization and generalization. Through extensive experiments spanning a wide range of model sizes and training data quantities, we uncover four distinct training dynamics, each arising from unique combinations of model size and training data quantity, formulating a theoretical framework for further analysis. Utilizing this framework, we establish connections between *double descent* and *grokking* and propose two verifiable predictions regarding the occurrence of *double descent*, both substantiated by our experimental results. Moreover, we expand our experiments to the multi-task learning paradigm, demonstrating how algorithm tasks can be turned into emergent abilities by mixing some pure memorization data. This offers a novel perspective to understand *emergent abilities* in Large Language Models.

## 1   Introduction

There are several interesting phenomena in Deep Learning, among which *grokking* (Power et al., 2022), *double descent* (Belkin et al., 2019; Nakkiran et al., 2020) and *emergent abilities* (Brown et al., 2020; Wei et al., 2022a; Ganguli et al., 2022; Srivastava et al., 2023) in current Large Language Models attract a lot of attention. Understanding these phenomena is important for us to reveal the mechanism of deep learning. Plenty of works (Liu et al., 2022; 2023; Thilak et al., 2022; Varma et al., 2023; Schaeffer et al., 2023; Michaud et al., 2023) have been done to explain these phenomenons from different perspectives. However, these works all concentrate on a single phenomenon and explain them separately. In this work, we provide a preliminary study to give a unified view of these three phenomena from the perspective of competition between memorization and generalization.

Our work is based on Varma et al. (2023)'s explanation for *grokking*. They attribute *grokking* to the competition between two distinct types of circuits in the model: one responsible for memorization, which achieves high training accuracy but poor validation accuracy, and another for generalization, capable of high performance in both training and validation. The latter, although slower to develop, proves more efficient in terms of parameter norms, leading to the model finally transferring from memorization to generalization to achieve higher efficiency. Intriguingly, the efficiency of the memorization circuit is inversely related to the volume of training data, indicating that larger datasets reduce its efficiency. In contrast, the efficiency of the generalization circuit remains consistently stable, regardless of the size of the training data. As a result, Varma et al. (2023) identified a critical dataset size, $D_{crit}$, delineating a boundary. Within this boundary, memorization and generalization circuits

---

* Corresponding author.

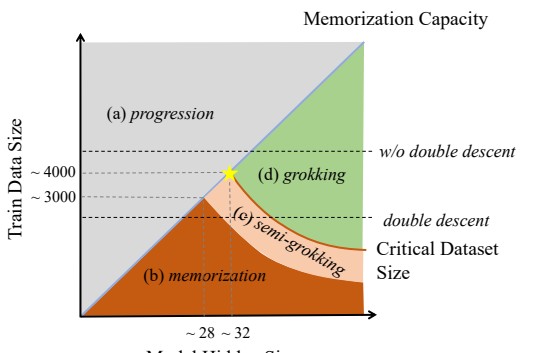
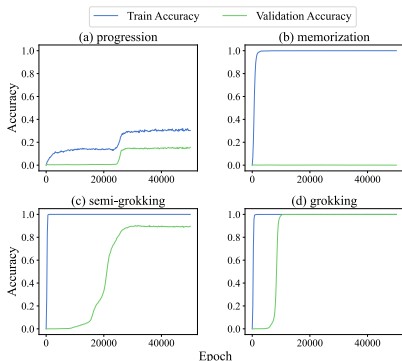

Figure 1: **Left:** The increasing memorization capacity and decreasing critical dataset size for larger models split the figure into four distinct zones including *progression*, *memorization*, *semi-grokking* and *grokking*. Each zone will show a specific training dynamic illustrated in the right side. Some important intersections are marked with estimated values. **Right:** Each figure represents a specific training dynamic: (a) *Progression*, demonstrated using a model with a hidden size of 8 and trained on 2600 data points. (b) *Memorization*, shown with a model having a hidden size of 32, trained on 2600 data points. (c) *Semi-Grokking*, depicted with a model of hidden size 56, trained on 2600 data points. (d) *Grokking*, visualized using a model with a larger hidden size of 56, trained on 4000 data points. These figures exemplify the variable training dynamics of models with different configurations to specific training data quantities.

demonstrate similar levels of efficiency and the model will show part of generalization ability. Crossing this threshold significantly enhances the probability of *grokking*.

Building upon their findings, our study expands the investigation to various model sizes. We firstly observe that smaller models require a larger critical dataset size for grokking, implying an inverse relationship between model size and the necessary amount of training data for grokking. Conversely, a model's memorization capacity is directly proportional to its size, indicating that smaller models have a reduced capacity for memorization. We confirm these relationships by experiments in § 2.

The reverse relationships with model size inevitably lead to an intersection point of the two curves (yellow star in Figure 1). The two curves with this intersection point create four distinct zones on the graph, each reflecting a unique training dynamic in our experiments as shown in the right side of Figure 1. (a) *Progression*: When the training data size is beyond model's memorization capacity, model will be unable to memorize all of them, causing model first memorizing as much training data as possible with a zero validation performance and then generalizing to part of validation data with an increase in train accuracy at the same time. (b) *Memorization*: For small amount of training data, model has the ability to memorize all of them and the memorization circuits is more efficiency than generalization. Therefore, model will only memorize the training data with zero validation performance. (c) *Semi-Grokking*: When the number of training data points approximates the critical dataset size, the model exhibits moderate generalization capabilities after memorizing all the training data. This behaviour was first identified by Varma et al. (2023). (d) *Grokking* (Power et al., 2022): When the number of training data is beyond critical dataset size, the generalization circuits become more efficiency than memorization, leading the model transfer from memorization to generalization long after training performance become perfect.

Analyzing Figure 1, we can easily link *double descent* with *grokking* and figure out when *double descent* will happen. Specifically, for training data volumes below the intersection point, the model undergoes *progression*, *memorization*, *semi-grokking* and then *grokking* with model size increasing, leading the validation performance first increases, then decreases, and ultimately increases again, which is exactly the *double descent* phenomenon (Belkin et al., 2019; Nakkiran et al., 2020). In contrast, when the quantity of training data surpasses the intersection point, an increase in model size results in the model through *progression* to *grokking*, resulting in a consistently positive correlation between model size and final validation performance. We

can see that the occurrence of *double descent* is highly related to the position of the intersection point. Therefore, to further verify our illustration, we conduct experiments to move the position of the intersection point and successfully transform a validation accuracy function without clear *double descent*, into one with obvious *double descent*. This transformation is achieved by shifting the critical dataset curve upward, thereby also shifting the intersection point towards the upper right.

Further, we extend our experiments to the multitask learning paradigm where an algorithm task and a pure memorization task are mixed to train the model. Interestingly, adding a pure memorization task largely hinders model from formalising the generalization circuits for the algorithm task. With model size increasing, model always achieves near zero validation performance until a relatively large model size, which is about 1570 times larger than training solely on the algorithm task. This phenomenon reminds us of the *emergent abilities* in Large Language Models (Brown et al., 2020; Wei et al., 2022a; Ganguli et al., 2022; Srivastava et al., 2023). The pretraining stage can also be seen as a multi-task learning process, where model has to remember numerous world knowledge while developing some general rules and abilities, such as reasoning. Our study suggests that this multi-task learning feature can be one important reason for the *emergent abilities* in LLM.

Overall, we make three key contributions in this study, which are outlined as follows:

- We introduce an innovative framework designed for analyzing the performance and training dynamics in consideration of both the size of the model and the quantity of training data.

- Utilizing this framework, we provide a nuanced illustration for the *double descent* phenomenon and establish a predictive method for identifying instances of *double descent* occurrence.

- By extending to multi-task learning which consists of algorithm and pure memorization tasks, we convert algorithm tasks into an emergent ability. This offers a novel angle for understanding *emergent abilities* in Large Language Models.

**Key Takeaways.** To better focus our readers' attention, we highlight the key takeaways from our analysis:

1. Critical dataset size for generalization decreases with model size increasing.

2. Memorization capacity increases with model size increasing.

3. The training dynamics can be categorized into four types.

4. *Progression* and *grokking* differ in parameter norm variations despite both showing delayed generalization ability during training.

5. *Double descent* is caused by training dynamics moving from *progression* to *memorization*, then to *grokking*.

6. We can make *double descent* more prominent by increasing the generalization difficulty.

7. *Emergent ability* can be introduced by mixture of memorization tasks and generalization tasks.

8. Separating memorization and generalization in parameter space leads to faster emergence.

## 2    Preliminary Study about Grokking

In this section, we first recall the preliminary setup of Grokking experiments in Varma et al. (2023), in which we can see the critical dataset size $D_{crit}^M$ as a function with model $M$ and extend the grokking experiments to different model sizes and training data size, resulting in an inverse relationship of critical dataset size $D_{crit}^M$ with model size, where larger models exhibit grokking with less training data.

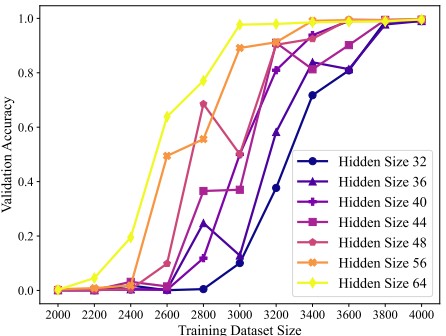 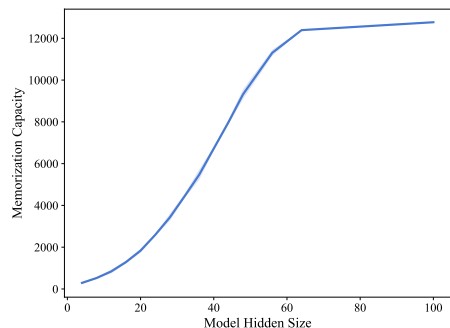

Figure 2: Final validation accuracy across various training dataset sizes and model hidden sizes. This figure demonstrates that larger models attain near-perfect validation accuracy with comparatively less training data, indicating a reduced critical dataset size for these models.

Figure 3: Graphical representation of the model's memorization capacity relative to its size. Each model is run with three distinct random seeds and the average performance is depicted. The light blue shaded region illustrates the 95% confidence interval.

## 2.1 Experiments Setup

**Task** Following Power et al. (2022) and Varma et al. (2023), we conduct experiments on the modular addition task for generalization without specific illustration.

$$(a + b) \bmod P; \text{ for } a, b \in (0, ..., P - 1) \text{ and } P = 113$$

By using different tuples of $a$ and $b$, this task can be easily split into non-overlap train and validation set. Therefore, the memorization behaviour and generalization behaviour can also be distinguished by the validation performance. Although Nanda et al. (2023) propose a method to identify the generalization circuit of modular addition, we directly use validation performance to distinguish for simplicity. We use $D_{add}^{train}$ and $D_{add}^{val}$ to represent the data number of training set and validation set on this modular addition task.

**Model** Following previous work (Nanda et al., 2023; Varma et al., 2023), we train a 1-layer simplified decoder-only transformer (Vaswani et al., 2017) model with 4 attention heads. We see the tasks above as classification tasks, where the label number is $P$. During training, a cross entropy loss $\mathcal{L}_{CE}$ with AdamW (Loshchilov & Hutter, 2019) optimizer is utilized. We mainly vary model size by adjusting its hidden size dimension $d_h$.

## 2.2 Grokking Experiments

In this study, we explore the impact of varying training dataset sizes, denoted as $D_{add}^{train}$, which range from 2,000 to 4,000 in increments of 200. Additionally, we investigate models with different hidden sizes, specifically $d_h \in \{32, 36, 40, 44, 48, 56, 64\}$. For each unique combination of dataset size and model hidden size, we perform experiments using 11 distinct random seeds, and their average performance is reported.

We first verify that modular addition task indeed show *grokking* in our experimental setup which is shown in Figure 1 (d). Model quickly achieves perfect training accuracy while validation accuracy takes more training steps to become perfect. By comparing Figure 1 (c) and (d), we can also find that model needs a specific amount of training data to show perfect generalization ability, which is the critical dataset size stated by Varma et al. (2023). Then, we present the final validation performance as a function of the training dataset size for different models in Figure 2 to analyse the impact of model size on critical dataset size.

TAKEAWAY-1 From the analysis presented in Figure 2, it is evident that models with larger hidden sizes tend to exhibit *grokking* (perfect validation performance) with smaller datasets. For instance, a model with a hidden size $d_h$ of 64 achieves near-perfect validation accuracy

with just $3,000$ training samples. In contrast, a model with a smaller hidden size of 32 requires an increase in training data size to $4,000$ to achieve similar validation accuracy. This trend is consistent across models with intermediate hidden sizes ranging between 32 and 64.

In addition, our research also verifies the phenomenon termed *semi-grokking* (Figure 1 (c)), as proposed by Varma et al. (2023). This phenomenon is very similar to *grokking* except for its partial generalization ability, which means model cannot show perfect validation performance in the end of training. Occasionally, *semi-grokking* can show several increasing stages of validation accuracy. Based on Varma et al. (2023)'s experiments, *semi-grokking* can be observed when the number of training data points, $D_{add}^{train}$, closely matches the critical dataset size for the current model, denoted as $D_{crit}^M$. Under these conditions, the model's memorization and generalization circuits demonstrate comparable efficiencies. This balance prevents the model from fully transitioning from memorization to generalization, resulting in *semi-grokking*.

As the results shown in Figure 2, a model with a hidden size $d_h$ of 64 shows middling validation accuracy and exhibits *semi-grokking* when the training dataset size, $D_{add}^{train}$, ranges between $2,000$ and $3,000$, where the critical dataset size, $D_{crit}^M$, for this model is approximately $3,000$. We observe that as the gap between $D_{add}^{train}$ and $D_{crit}^M$ narrows, there is a general improvement in the model's generalization ability, as reflected in its final validation accuracy.

## 2.3 Memorization Experiments

Since *grokking* requires the model to memorize all of the training data firstly, we designed experiments that specifically gauge the model's ability to memorize. For this purpose, we assign random labels to the training data (Zhang et al., 2021; Doshi et al., 2024), compelling the model to focus solely on memorization. Given the modular addition task's adherence to the commutative law, which could affect the model's memorization capability on this task[1], we assign identical random labels to $(a + b) \bmod P$ and $(b + a) \bmod P$. This approach ensures a more accurate assessment of the model's true memorization ability for modular addition task. Our experiments encompass all $12,769$ pairs as training data, and we evaluate the model's accuracy on this entire set to represent its memorization capacity.

TAKEAWAY-2 Unsurprisingly, the findings, illustrated in Figure 3, reveal a distinctly positive correlation between the model size and its memorization capacity; larger models are capable of memorizing more training data.

# 3 Proposed Framework

In this section, we introduce our framework designed to analyze both the training dynamics and final validation performance. This approach is grounded on two key assumptions, which are substantiated by the experiments detailed in § 2.

**Assumption 3.1.** The critical dataset size for a model $M$ ($D_{crit}^M$) is negatively correlated to the model's size. This implies that larger models require less data to exhibit *grokking*.

**Assumption 3.2.** The memorization capacity of a model $M$ ($D_{mem}^M$) correlates positively with the model's size, indicating that larger models have a greater capacity to memorize training data.

TAKEAWAY-3 Drawing from Assumption 3.1 and Assumption 3.2, we can construct a graphical representation for a specific task, as illustrated in Figure 1. This graph delineates the relationship between memorization capacity and critical dataset size, highlighting their intersection point (marked by a yellow star in Figure 1). Consequently, this demarcates

---

[1]It is easier for model to memorize training data which obeys commutative law, since it only needs to memorize half of the training data.

four distinct zones, each representing a unique training dynamic. These dynamics will be discussed detailly in the subsequent discussion (right side of Figure 1).

**Progression** In scenarios where the training dataset $D^{train}$ surpasses the memorization capacity $D^{M}_{mem}$ of a model $M$, the model is incapable of memorizing the entire dataset. This limitation leads to a two-stage learning dynamic. Initially, the model memorizes a portion of the training data, which does not translate into improved validation performance. Subsequently, the model begins to generalize, improving both its training accuracy and validation performance, as shown in Figure 1 (a).

**Memorization** When the training data $D^{train}$ falls below the memorization capacity $D^{M}_{mem}$ of the model, it can memorize the entire dataset. However, if $D^{train}$ is also significantly less than the critical dataset size $D^{M}_{crit}$, memorization circuits outperform generalization circuits in efficiency. This leads the model to opt for pure memorization, resulting in negligible validation performance, as illustrated in Figure 1 (b).

**Semi-Grokking** In cases where the training data $D^{train}$ approximates the critical dataset size $D^{M}_{crit}$, the efficiency of memorization and generalization circuits becomes comparable. This parity causes the model to struggle with transitioning entirely to generalization circuits. Consequently, the final model is a combination of both memorization and generalization circuits, yielding moderate validation accuracy. Additionally, multiple stages of increase in validation performance may be observed, indicating repeated shifts between memorization and generalization circuits, as shown in Figure 1 (c).

**Grokking** In the situation where the training data $D^{train}$ is between the memorization capacity $D^{M}_{mem}$ and the critical dataset size $D^{M}_{crit}$, the model is able to memorize the entire training dataset. However, the circuits dedicated to memorization are less efficient compared to those for generalization. Initially, the model achieves perfect training accuracy through memorization. It then transitions to generalization circuits for higher efficiency, leading to a training dynamic demonstrated in Figure 1 (d), where validation performance reaches near perfection well after the model has overfitted to the training set.

**Discussion** Despite *progression* and *grokking* show similar validation performance dynamic during training, it is crucial to note the distinction between them in two fundamental aspects. Firstly, the generalization circuits in *progression* show up a moderate level of training accuracy. In contrast, *grokking* is characterized by the formation of generalization circuits only after the model has achieved perfect training accuracy. Secondly, the underlying mechanisms driving the formation of generalization circuits differ significantly. In the case of *progression*, generalization circuits are induced by the constraints of cross-entropy loss, as the model is unable to completely memorize all training data, preventing it from reaching near-zero training loss. Consequently, the generalization circuits in *progression* are developed to minimize training loss while concurrently enhancing validation performance. However, the emergence of generalization circuits in *grokking* is attributed to the model's preference towards more efficient circuits, which is minimizing the model's parameter norm. This leads to a notable trend for generalization where, during *grokking*, the model's parameter norm tends to decrease, whereas in *progression*, an increase in the parameter norm is typically observed, which is shown in Figure 7. (TAKEAWAY-4)

## 4 Illustrate Double Descent

The phenomenon of *double descent*, as observed by Nakkiran et al. (2020), reveals an intriguing pattern wherein the increase in model size firstly detrimentally impacts validation performance before finally contribute positively to the validation performance. In this section, we provide a detailed exploration of the *double descent* phenomenon, coupled with predictions regarding its occurrence given our framework in § 3. Subsequently, we undertake a series of experiments designed to validate our theoretical illustration.

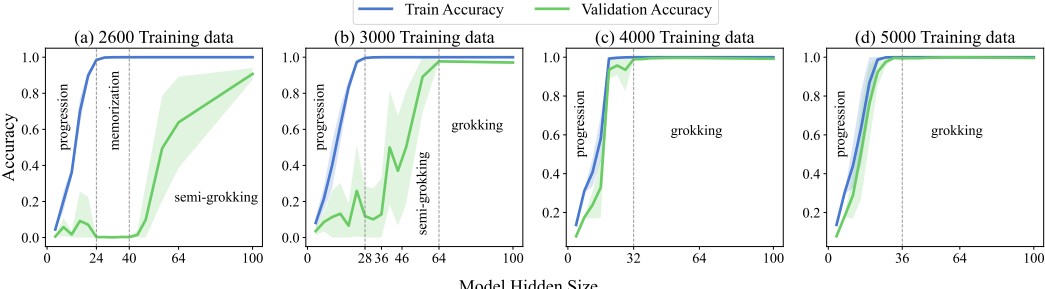

Figure 4: Results on modular addition task with varying training data sizes. Each experiment is conducted 11 times using distinct random seeds, and mean accuracy is reported. The light-colored regions denote 95% confidence intervals. Consistent with our expectations, smaller amount of training data tends to induce the *double descent* phenomenon.

## 4.1 Illustration About Double Descent

TAKEAWAY-5 Our framework elucidates that the phenomenon of *double descent* is likely to manifest when the number of training data, $D^{train}$, is substantially lower than the intersection point of two curves in Figure 1. In such scenarios, the model undergoes a series of stages: *progression*, *memorization*, *semi-grokking*, and finally *grokking*. Initially, as the model size increases during the *progression* phase, there is an enhancement in its generalization ability. However, this ability declines, potentially to zero, upon transitioning into the *memorization* stage. This decline is in alignment with the critical interval of model size postulated by Nakkiran et al. (2020). As the model continues to grow, it enters the *semi-grokking* and *grokking* phases, where its generalization ability is revived, leading to a secondary increase. This relationship results in the *double descent* curve observed in validation performance[2]. Conversely, in instances where the training data exceeds the intersection point, the model predominantly experiences *progression* and *grokking*, resulting in a consistent upsurge in generalization ability as the model size increases. Under these conditions, the *double descent* phenomenon does not occur.

## 4.2 Experiments

To validate our illustration, we carried out a series of experiments centered on the modular addition task, as detailed in § 2. We set the training data sizes to $\{2600, 3000, 4000, 5000\}$ and varied the model's hidden size from 4 to 100, following the sequence $\{4, 8, 12, 16, 20, 24, 28, 32, 36, 40, 44, 48, 56, 64, 100\}$. Each configuration was tested across 11 experiments with distinct random seeds, and the average performance was reported, accompanied by a 95% confidence interval. These results are depicted in Figure 4.

Observations from Figure 4 (a) reveal that with a training dataset of 2600 samples, the models transition through the stages of *progression*, *memorization*, *semi-grokking*[3]. In the *progression* phases, larger models generally demonstrate improved validation performance. However, the *memorization* phase, evident when the model's hidden size ranges between 24 and 40, leads to zero validation performance, resulting in the *double descent* pattern for this dataset size.

Increasing the training data slightly to 3000 samples eliminates the *memorization* stage. The models then progress through *progression, semi-grokking*, and *grokking*, resulting in a less pronounced *double descent* curve. Still, a dip in validation performance is noticeable for model sizes between 28 and 36, as shown in Figure 4 (b). Outside this range, there is a general trend of increasing performance with larger model sizes.

---

[2]The "descent" in double descent refers to validation error/loss, therefore in terms of validation performance, it's validated by the two "increase" stage.

[3]Due to the small number of training data, *grokking* doesn't happen in this model size range.

With the training data further increased to 4000 and 5000, models bypass the *memorization* and *semi-grokking* stages, transitioning directly from *progression* to *grokking*. This leads to a consistent improvement in validation performance as model size increases, without any occurrence of *double descent*, as illustrated in Figure 4 (c) and (d).

### 4.3   Make Double Descent More Prominent

TAKEAWAY-6 Under our proposed framework, the occurrence of *double descent* is highly related to the position of intersection point in Figure 1. Therefore, moving in the intersection point towards the upper right results in an expanded range of training data sizes, during which model will show pronounced *double descent*. Consequently, a specific quantity of training data that previously does not demonstrate *double descent* can be transformed into one that does. To move intersection point towards the upper right, what we need is to increase the difficulty of generalization. To validate this hypothesis, we designed experiments centered around a task more complex than the modular addition which is defined as:

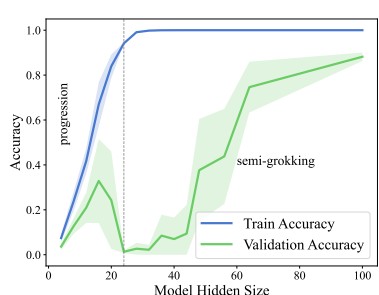

Figure 5:   Experimental results of $(a + b)^2$ mod $P$ with 3000 training data points.

$$(a + b)^2 \bmod P; \text{ for } a, \ b \ \in (0, ..., P - 1) \text{ and } P = 113$$

Given that this task also adheres to the commutative law, it does not affect the memorization capacity curve. For a direct comparison, we use 3000 training data, the same size that did not exhibit a very clear *double descent* in Figure 4 (b). We keep all other experimental setup the same as modular addition task. The outcomes are depicted in Figure 5.

The results from this more complex task reveal a more pronounced *double descent* phenomenon in the validation performance compared to the one observed in Figure 4 (b), which utilized an identical quantity of training data. Notably, in this task, the model exhibits *memorization* stage with zero validation performance when the hidden size of the model is 24. This *memorization* stage is not observed in the modular addition task with the same training data size of 3000. These behaviours corroborate our previously stated hypothesis and verify our illustration about *double descent*.

## 5   Multi-Task Learning Leads to Emergent Ability

In this section, we expand our research to the multi-task learning paradigm, which combines modular addition task, with a task focused solely on memorization. This approach reveals that the model's generalization ability on the algorithm task remains negligible until the model reaches a substantially larger size—specifically, 1570 times larger than training on a single task. As a result, the validation performance on algorithm task show an emergent phenomenon relative to model size.

**Experiment Setup**   Our experiments utilize the modular addition task as the generalization component. For the memorization task, we assign random labels to a subtraction task, compelling the model to memorize these associations. By incorporating different calculation symbols $(+, -)$ into the input tokens, we ensure that the memorization and algorithm task have no overlapping inputs. Our experiments involve 3000 data points from the modular addition task and a varying number of memorization data points, ranging from 3000 to 6000. We adjust the model size from a 1-layer transformer with a hidden size of 64 to an 8-layer transformer with a hidden size of 1024. Each experiment is conducted three times, and the highest validation accuracy is reported to showcase each model's optimal generalization ability. The results are presented in Figure 6.

TAKEAWAY-7 We observe that incorporating pure memorization data significantly impedes smaller models in developing generalization circuits. The emergence of generalization ability in the modular addition task is notable, with models typically displaying substantial validation performance at relatively larger sizes, approximately 1570 times larger than those trained solely on the modular addition task. Additionally, the volume of memorization data appears to have minimal impact on the emergent model size.

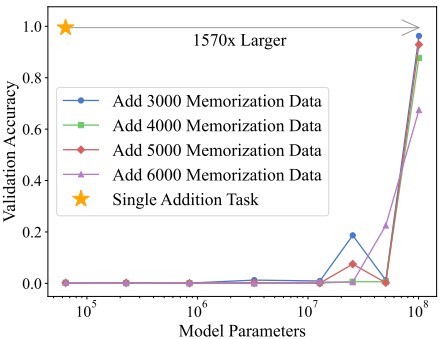

Figure 6: Adding a pure memorization task into the modular addition task makes it become an emergent ability.

**Discussion** From the perspective of the competition between memorization and generalization, the presence of a pure memorization task prevents the model from totally transitioning from memorization to generalization after memorizing all training data, as there are no generalization methods for the pure memorization task. However, once a sufficiently large model size is reached, the model's memorization capacity significantly exceeds the training data volume, allowing it to show functional differentiation for different abilities including memorization and generalization. The experiment in Appendix E, which reduces the emergent model size by allocating separate parameters for generalization and memorization data, further supports this hypothesis. This phenomenon echoes the *emergent abilities* observed in current Large Language Models, since the pretraining stage can also be seen as a multi-task learning scenario, where the model must retain a vast array of world knowledge while acquiring general rules and capabilities, such as in context learning (Brown et al., 2020) and multi-step reasoning (Wei et al., 2022b). This observation also elucidates the hypothesis proposed by Hu et al. (2024), where it is hypothesized that emergent abilities are formed through the competition of different neural circuits. Although the emergence of abilities in current LLMs is driven by a multitude of complex factors, we believe our experiments will offer fresh insights into understanding *emergent abilities* in LLMs and stimulate further research in this area.

## 6 Related Work

**Grokking** The phenomenon of *grokking*, where models demonstrate exceptional generalization capabilities well beyond the point of overfitting to training data, was first identified by Power et al. (2022) in various algorithm tasks. Thilak et al. (2022) demonstrate that *grokking* often comes with "Slingshot Effects," which may play a pivotal role in its emergence. Delving into the underlying mechanisms, Liu et al. (2022) approached *grokking* from a representation learning standpoint, uncovering its association with structured representation development. Merrill et al. (2023) analyse *grokking* through the competition between subnetworks. Beyond algorithm tasks, recent work (Liu et al., 2023; Murty et al., 2023) also discovered that *grokking* happens in a broader spectrum of realistic tasks. Additionally, Junior et al. (2024) explored predictive markers of grokking, highlighting "oscillations" within the loss landscape as potential indicators. A novel perspective by Varma et al. (2023) suggests that grokking can be interpreted through the competition between memorization and generalization circuits, influenced by the efficiency of these circuits. This demonstration can be included in our framework by a vertical line on the right of the intersection point in Figure 1.

**Double Descent** The concept of *double descent*, as introduced by Belkin et al. (2019), illustrates a unique pattern in model validation error: an initial decrease, followed by an increase, and then a subsequent decrease, in correlation with the growing size of the model. The increase of validation error typically coincides with the model's training error nearing zero. Expanding upon this concept, Nakkiran et al. (2020) conducted a

comprehensive examination of the double descent phenomenon across varying model architectures, datasets, and optimization techniques. In a parallel effort, Davies et al. (2022) attempted to bridge the concepts of *double descent* and *grokking*. This was approached through a proposed duality between model size and scaling time, though this hypothesis remains to be empirically verified.

**Emergent Abilities**    The concept of *emergent abilities* has garnered significant interest in the era of the development of LLMs. Wei et al. (2022a) provide a thorough investigation into different abilities across various models, characterizing them as capabilities that are absent in smaller models but suddenly present in larger ones. Barak et al. (2022) gave a comprehensive analysis on a synthetic task about its emergence. Caballero et al. (2023) introduced a complex function, modeled on a piece-wise power law, to encapsulate various phenomena, including *emergent abilities*. Schaeffer et al. (2023) attributed the emergence of these abilities to the non-smooth metrics employed in task evaluation. Taking a predictive angle, Hu et al. (2024) succeeded in forecasting certain *emergent abilities* by employing a metric with infinite resolution. Different from these works, our research delves into this phenomenon from a unique angle, focusing on the competition between memorization and generalization.

## 7    Conclusion

In this paper, we conduct a comprehensive study on algorithm task to show varying training dynamics across different model sizes and training dataset sizes and build a framework to analyse different phenomena. Based on this, we offer a comprehensive illustration of *double descent* and establish its connection with *grokking*. By integrating memorization and generalization tasks, we successfully induce emergent behaviour into generalization tasks, shedding new light on the understanding of *emergent abilities*. Future work on more realistic tasks and models will be crucial for a deeper and more comprehensive understanding of deep learning mechanisms.

## Acknowledgements

This work is supported by the National Science and Technology Major Project (2020AAA0106502), National Natural Science Foundation of China (No, 62236011) and Institute Guo Qiang at Tsinghua University.

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

# A    Limitations

Our experiments and analysis are all conducted on algorithm tasks like modular addition due to their simplicity. Since there is no noise data and no overlap between training and validation set, we can easily distinguish model's memorization and generalization behaviours, and therefore fully focus on model's generalization ability in this task. However, we believe our framework can also make some sense for more realistic setting and real tasks since the concept of memorization and generalization still hold under these circumstances. Further experiments on realistic setting will be an important future direction.

Besides, we only consider pure memorization and generalization in this work. However, neural networks can encompass different circuit types that may also affect the model's ability to memorize and generalize. For instance, the communicative law in algorithmic tasks can introduce heuristic circuits that enhance the model's capacity to memorize training data. Although we have taken steps to eliminate such heuristic bias in our training data, other biases may still exist that we have not accounted for.

# B    Training Details

**Data Split**   Since all the algorithm tasks in this paper follow communicative law, we split train and validation data based on tuples to avoid heuristic generalization method, which means if $(a + b) \bmod P$ in training set, then $(b + a) \bmod P$ is also in training set. This is also the reason that communicative law can influence model's memorization capacity since it only needs to memorize half of the training data.

**Model Structure**   We utilize a 1-layer simplified decoder-only transformer with 4 attention heads for most of our experiments, which don't have bias in linear layer and layernorm. We utilize ReLU as the activation function in MLP and the intermediate size of MLP is 4 times model's hidden size (dimension of model's embedding). We mainly modify model's hidden size without changing attention heads number to change model size except for experiments in § 5 and Appendix E, where we also increase the layer number to support larger models.

**Training Hyper-Parameters**   We implement our experiments based on the trainer of Huggingface transformers (Wolf et al., 2020). During training, we utilize a dropout of 0.1 and a constant learning rate of 0.001. AdamW with weight decay of 1.0 is utilized for optimization. For larger models experiments in § 5 and Appendix E, we find high learning rate of 0.001 will cause the model even cannot memorize the training data, so we adjust the learning rate to 0.0005.

# C    Parameter Norm Variations on Progression and Grokking

As we have discussed in § 3, the concepts of *progression* and *grokking* differ in two fundamental ways. The first difference lies in the training accuracy at the point of generalization, a distinction that is clearly observable in Figure 1 and Figure 7. The second difference pertains to the underlying reasons for the emergence of generalization abilities. In the case of *progression*, generalization is triggered by non-zero training loss by pure memorization, leading to a reduction in both training and validation losses by generalization. Conversely, *grokking* occurs when generalization circuits prove to be more efficient than those used for memorization, a phenomenon that is verified by Varma et al. (2023). This distinction can also be highlighted by the observed trends in parameter norms: during the shift from memorization to generalization, *grokking* exhibits a decreasing parameter norm, whereas *progression* demonstrates an increasing parameter norm, as depicted in Figure 7.

# D    Double Descent with Full Test Set

Different from modular addition task, which we can split into non-overlap training and validation set, real tasks always show some overlap or similar data between training and

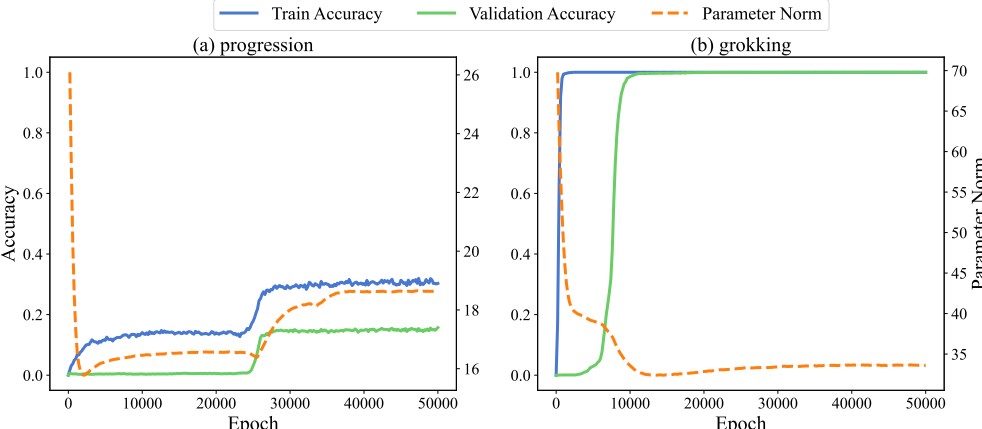

Figure 7: Difference of parameter norm variations during training for *progression* and *grokking* in Figure 1. Notably, the parameter norm in *progression* exhibits a marked increase when the transition to generalization happens. Conversely, in *grokking*, the parameter norm demonstrates a decreasing trend as generalization ability is formalized, highlighting the difference mechanisms driving generalization in each case.

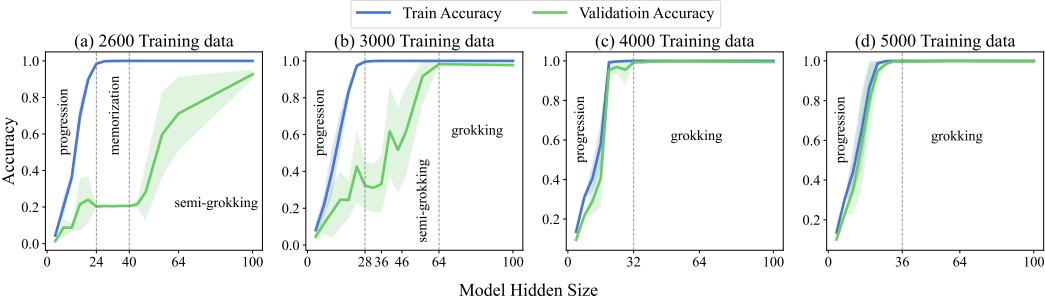

Figure 8: Experimental results on the modular addition task with varying training data sizes. **The validation set is all the data pairs in this task**. Each experiment is conducted 11 times using distinct random seeds, and the mean accuracy is reported. The light-colored regions denote 95% confidence intervals. Similar to the Figure 4, 2600 training data and 3000 training data show *double descent*.

validation set (Lewis et al., 2021), where memorization of the training data will lead to increased performance in validation. Therefore, we further examine model's performance on all the data pairs of modular addition task which include the training data and unseen validation data to measure model's ability (no matter memorization or generalization) on this task. The results are shown in Figure 8. We can see that model show more significant *double descent* with 2600 and 3000 training data compared with Figure 4 (a) and (b). Besides, 4000 and 5000 training data still don't show *double descent* in this scenario, which also verifies our illustration about *double descent* in § 4.

## E   Further Experiments on Emergent Ability

TAKEAWAY-8 To enhance our understanding of how combining a pure memorization task with a modular addition task results in emergent ability, we hypothesize that the inclusion of a pure memorization task inhibits the model's shift from memorization to generalization. This occurs because there are no generalization methods specifically for the pure memorization task. To investigate this further, we designed experiments to segregate parameters dedicated to memorization and generalization task. Prior studies suggest that the feed-

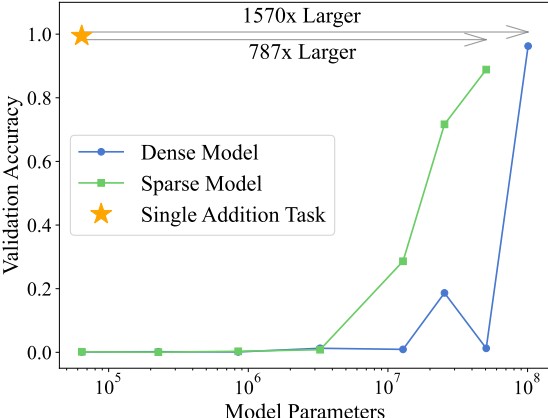

Figure 9: Experiments on multi-task learning with a pure memorization task and modular addition task. This study delves into the effects of manually constructing a sparse model to separately address pure memorization and modular addition data. Through this approach, we significantly accelerate the emergence of the modular addition capability in terms of model size. This finding highlights the crucial role of functional differentiation within neural models in fostering the emergence of new abilities.

forward layer in the Transformer architecture predominantly facilitates memorization (Geva et al., 2021). Additionally, this ffn layer in transformer is also crucial for the generalization process in the modular addition task (Nanda et al., 2023; Chughtai et al., 2023). Therefore, we partitioned the feed-forward layer into two specialized sections by dividing the intermediate dimension, akin to the current MoE architecture (Lepikhin et al., 2021; Fedus et al., 2022). In this setup, one part exclusively processes the modular addition task data, while the other focuses solely on the memorization task data. Our experiments, conducted on 3000 instances of modular addition and 3000 memorization data, are depicted in Figure 9. The results demonstrate that manually creating a sparse network significantly accelerates emergence of the modular addition task capabilities with the same training dataset. This finding underscores the critical role of functional differentiation and sparseness (Zhang et al., 2022) in language models for developing *emergent abilities*.

