# OpenReview forum: "Unified View of Grokking, Double Descent and Emergent Abilities: A Comprehensive Study on Algorithm Task"
_colmweb.org/COLM/2024/Conference — COLM_

### Official Review · Reviewer_ANPL · 2024-04-25

**Rating:** 6
**Confidence:** 3
**Ethics Flag:** 1

**Summary:**

This paper considers some of the most intriguing phenomena in the field (grokking, emergence, double descent), which have proven to be interesting yet poorly understood. The most direct contributions are some interesting findings that arise from systematic exploration of different model and data sizes.

In short:
1. Methodologically, I do not find the paper to make meaningful contributions. The core ideas has been identified in prior work, and the extension of sweeping systematically over model/data sizes is extremely natural. This is not to say the paper needs to provide methodological insight to be valuable, just that I don't see it as providing methodological insight.
2. With respect to rigor, I generally find the decisions and execution reasonably, though I did not interrogate them too deeply.
3. With respect to findings, I see this as the area where the paper shines. The figures, and the discussion of them, seem clearly of interest and make progress in advancing our understanding of these fascinating topics.

So, on balance, I generally am positive on the paper, find that it certainly befits COLM, and think it will gain traction as an interesting paper that makes people think and galvanizes further research.

**Questions To Authors:**

No specific questions - I will monitor the responses to other reviewers to see if I want to adjust my score.

If you want to best attempt to raise my score:
1. Convince me the framework is the right conceptual model
2. Tell me how you will rewrite parts to make the writing sharper

There are a lot of terms/concepts/ideas floating around in the paper. I would encourage adding some clear takeaways after each section or so on. If you want the reader to reminder some findings, make them more salient (in intro, in the actual content sections), because I would worry readers will sort of leave this paper with a muddled conceptual model of what it says a few days after reading it.

**Reasons To Accept:**

The main reason to accept the paper is that it advances our understanding of emergence/grokking, both in the process-level sense of conducting useful new experiments and in the outcome-level sense of providing new qualitative insights.

The authors present their framework as a key contribution. I am not sure I would have foregrounded the framework so much, since it is hard to see why this is the right/preferable way to conceptualize what is going on, but it seems reasonable enough. I do not work in this area, so I will defer to other reviewers/the community in the future to see if this framework is worthy of uptake, but for now I see at as a second-order reason to be supportive of acceptance.

**Reasons To Reject:**

Given the emphasis placed on the framework, I am generally left dissatisfied: how do we arrive at this framework, why is it good, and how would it connect to a more theoretical account? I am not even sure it makes sense to call it a framework: it seems more reasonable to say it is a way of assigning terms to some phenomena, which perhaps helps build common terminology, but I am not sure this really advanced my conceptual understanding.

While the paper is clear in a certain structural sentence, I found the writing more confusing than helpful in most places, perhaps because it is meandering. Since this is a general comment, it is hard to isolate specific paragraphs that could benefit from re-writing, so I would say the key principle I feel is not upheld is each sentence does not have a clear rhetorical purpose, nor is the ordering of the sentences particularly deliberate as far as I can tell. As a result, while I don't object to the writing, I feel it will not only inhibit my understanding but that of other readers.

---

> ### Author Rebuttal · Authors · 2024-05-31
>
> Thank you for the valuable comments that help us improve our work. Here is our response:
> > Questions about framework.
>
> 1. How do we arrive at this framework?
>
>    We developed this framework based on two key experimental findings: the decreasing curve of critical dataset size for generalization and the increasing curve of the model's memorization capacity, as shown in our *Preliminary Study*.
> 2. why is it good?
>
>    We believe the framework has two advantages:
>
>    Firstly, it is consistent with existing work and links various phenomena.  For example, a vertical line in the right matches Varma's results[1]. This framework can demonstrate the relationship between double descent and grokking. We can imagine that if the critical dataset curve also increases with model size, then double descent would not occur, which violate double descent theory.
>
>    Secondly, a good framework helps predict model's performance. In Section 4.3, we make predictions based on it that increasing difficulty of generalization would result in a more pronounced double descent, which was also confirmed by our experiments.
> 3. How would it connect to a more theoretical account?
>
>    In the paper, we analyzed the framework through the lens of competition between memorization and generalization. Here, we provide an optimization-based analysis.
>
>    With the AdamW optimizer, both the gradient from cross-entropy loss and weight decay influence optimization. The memorization capacity curve relates to the loss, aiming for near-zero training loss. Beyond this curve, non-zero loss pushes the model toward generalization. Below the curve, near-zero loss highlights the impact of weight decay. The critical dataset size curve represents where weight decay doesn't favor memorization or generalization. Thus, different training dynamics also reflect the varying effects of cross-entropy loss and weight decay.
> > make the writing sharper
>
> We also noticed similar issues in the current content due to the numerous findings and experimental results. To address this, we will add key takeaways at the end of the introduction, directly linking to the corresponding sections. Here are some examples of takeaways:
> 1. Critical dataset size for generalization decreases with model size. -> Sec 2.2
> 2. Memorization capacity increases with model size. -> Sec 2.3
> 3. The training dynamics can be categorized into four types. -> Sec 3
>
> [1] Explaining grokking through circuit efficiency. Varma et al.

---

> > ### Comment · Reviewer_ANPL · 2024-06-04
> > **Reviewer response to rebuttal**
> >
> > Thanks for the response, I have read the discussion with other reviewers as well and will keep my score as-is and recommend acceptance to the AC.

---

### Official Review · Reviewer_R1E7 · 2024-05-10

**Rating:** 8
**Confidence:** 3
**Ethics Flag:** 1

**Summary:**

This paper aims to analyze observations regarding training dynamics in neural network-based models (grokking, double descent, emergent abilities) under a unified lens of memorization/generalization competition, following the Varma et al.'s explanation of grokking. Specifically, the claim is that the interaction between two factors (inverse correlation between scale and critical dataset size for grokking, correlation between scale and memorization capacity) can provide an explanation for all of the aforementioned phenomena.

* Quality: The hypothesis being tested is clearly stated and the experiments conducted to test it seem sound.
* Clarity: The exposition and writing are generally easy to follow, barring a few minor suggestions to improve (see question to authors section for more details).
* Originality: The attempt to tie together the various observations about training and scaling dynamics in recent literature is novel. However, this is not my direct area of expertise so I haven't been following the relevant literature fully---I am basing my assessment based on the claimed contribution.
* Significance: The work makes a "connecting the dots" type of contribution that can potentially generate a wide range of concrete predictions about training and scaling dynamics of Transformer models. Generalizability of the findings outside the exact setup (architecture, task) tested is limited, but the hypothesis and predictions laid out are concrete enough that their generalizability can be easily tested under more diverse settings.

**Questions To Authors:**

- I think the patterns visualized in Figure 4 doesn't itself serve as full evidence that the patterns observed during the course of training is exactly "grokking" or other phases---for example, the part of the graphs labeled "grokking", because in the graph itself just shows the endpoint of training (I assume), could equally be derived from a set of underlying models with training dynamics showing a fully overlapping train/val graphs from the beginning to end, with endpoints converging to full accuracy. I'm not saying this is the case for the results reported and I do believe that if we zoomed in to the individual models' training dynamics, they are probably adhering to the claimed patterns. I'm just saying that the visualizations from the Figure 4 themselves don't inherent imply/serve as evidence that the labels they are being assigned unless one actually looks at the full training dynamics for the individual cases, which might mean that a different set of metrics that can show signatures of the phases need to be devised in addition to simply visualizing the final training/val performance across varying model size.
- What did the performance on the memorization examples in the multitask experiment look like?
- I would also be interested in knowing what the lower bound of the memorization dataset size for observing the effect in Section 5 - the smallest size (3000) is still quite substantial since it matches the size of the modular addition task dataset. It would be very interesting if we observed such an effect with only a very small number of memorization examples.

More minor presentation suggestions:
- Currently the right panel of Figure 1 is serving a dual purpose (illustrative and also showing experimental results), and they seem to contain insufficient information to actually show the points of the experiments. For example, the text says "We first verify that modular addition task indeed show grokking in our experimental setup which is shown in Figure 1 (d).", but it is unclear what subset of the experiments that Figure (1d) is pointing to (especially in contrast to 1c---the details are only vaguely suggested: "model needs specific amount of training data")
- Typo: Memorizatioin -> Memorization
- (1c) doesn't seem to show "multiple plateaus" (at least in a way that significantly differs from plateaus in 1d)

**Reasons To Accept:**

This is a clearly written, interesting paper proposing a unifying explanation for various prior observations about training and scaling dynamics.

**Reasons To Reject:**

Limited testing environments (architectural and task variation), but I don't think this is a strong reason to reject.

---

> ### Author Rebuttal · Authors · 2024-05-31
>
> Thank you for the valuable comments that help us improve our work. Here is our response:
>
> > Tag in Figure 4
>
> We acknowledge that final performance does not fully support the labels for different stages, especially grokking. Visualizing both final performance and training dynamics in a single figure is challenging. Thus, we propose another metric for measuring delayed generalization during grokking. We define $S_t(M)$ as the training steps to achieve 99% of maximum training accuracy and $S_v(M)$ for validation accuracy. The difference, $S_{delay}(M) = S_v(M) - S_t(M)$, quantifies the delayed generalization. A large $S_{delay}(M)$ indicates grokking. Below are the $S_{delay}$ values for several *grokking* experiments in Figure 4:
> | Training Data | Hidden Size | $S_{delay}$ |
> | - | - | - |
> | 3000 | 64 | 27,022 |
> | 3000 | 100 | 19,704 |
> | 4000 | 64 | 7,371 |
> | 4000 | 100 | 7,302 |
>
> All models exhibit delayed generalization. As the model size and data size increase, the number of delayed steps for generalization decreases significantly.
>
> > The performance on the memorization examples in the multitask experiment
>
> In all our multitask experiments, the performance on memorization examples quickly achieved 100%.
>
> > lower bound of the memorization dataset size for observing the effect in Section 5
>
> We conducted additional experiments with fewer memorization data points. The results are shown below, where the columns represent model size, and the rows represent number of memorization data:
> |   | 847K | 3.3M | 12.8M | 25.4M | 50.6M | 101M |
> | - | - | - | - | - | - | - |
> | 3000 | 0 | 1 | 1 | 19 | 1 | 96 |
> | 2000 | 0 | 0 | 0 | 0 | 3 | 97 |
> | 1000 | 7 | 69 | -  | - | - | -
> | 500 | 71 | 91 | - | - | - | - |
> | 100 | 91 | 96 | - | - | - | - |
>
> The impact of memorization data decreases with fewer data points, which is expected. A small amount of memorization data acts as noise, and the model can typically ignore small noise and identify correct patterns. These results also suggest a lower bound for memorization data required to exhibit emergence lies between 1000 and 2000 points when using 3000 generalization data.
>
> > Experimental setting in Figure 1
>
> We will add a brief explanation in the main article to provide better clarity. Currently, the specific experiment settings are described in the caption of Figure 1.
>
> > (1c) doesn't seem to show "multiple plateaus"
>
> We will rephrase it as "multiple stages of increase", as shown in the curve around 20,000 steps in Figure 1(c).

---

> > ### Comment · Reviewer_R1E7 · 2024-06-04
> >
> > Thanks for the response and the additional experiments! My rating was positive to begin with and I didn't find glaring issues from other reviews, so I'll maintain my original score.

---

### Official Review · Reviewer_q1HD · 2024-05-18

**Rating:** 8
**Confidence:** 3
**Ethics Flag:** 1

**Summary:**

This paper proposes a hypothesis on the relationship between dataset size, the number of parameters in a neural model, and the grokking phenomenon. The authors propose an account of four distinct training dynamics based in this explanation, and show empirical evidence verifying each prediction. There is also an exploration of the connection between these results to double descent losses, as well as emergent abilities during pretraining.

I found the paper mostly easy to follow. While the set of experiments at first seems like Varma et al. (2023) with the added variable of model size, I think the proposal that double descent and emergent abilities can be explained by the proposed model is ambitious and interesting.

**Questions To Authors:**

**Suggestions**:
1. Fig. 3 shows an interesting non-linear shape. This would be interesting to discuss; what kind of relationship (e.g., logistic, logarithmic) do we expect between hidden size and memorization capacity? Is this really a relationship between number of parameters and memorization, or more of a relationship with width specifically?
2. In Figure 4, the legend should go outside the figures.
3. In general, it would be nice to use fewer evaluative words/phrases when describing the proposed work. For example, “fresh insights”, “successfully”

**Typos**:
- p.5: Memorizatioin -> Memorization
- p.7: “into one does” -> “into one that does”

**Reasons To Accept:**

1. Thorough empirical validation of a series of clearly defined hypotheses.
2. The experiments are well-controlled.
3. The topic seems impactful. The paper focuses on presenting a unified account of various phenomena that are hypothesized to underlie the impressive capabilities of LLMs.

**Reasons To Reject:**

1. Could engage with the relevant literature much more thoroughly. For example, [1,2,3] should be cited when discussing grokking and emergent abilities. [4] should be cited when discussing how larger models have greater memorization capacity/memorize more quickly. The results should be contextualized with [5].
2. The way Section 5 is written, it sounds like “implicit multi-task learning setups during pretraining are the cause of emergent abilities.” I think a much clearer framing would be something more like “this is actually a case of delayed grokking that occurs when the training corpus can only be explained by a mixture of both memorization *and* generalization circuits.” Also, this may just be one variant of emergent abilities; perhaps more complex latent linguistic phenomena simply require more parameters to reliably acquire.
3. One could view this as a simple extension of Varma et al. (2023); the main empirical contribution seems to be the addition of the model size variable. I personally don’t find this to be a significant downside, given the interesting perspectives presented and the decently thorough set of experiments.

References:

[1] Merrill et al. (2023). A Tale of Two Circuits: Grokking as Competition of Sparse and Dense Subnetworks. https://openreview.net/pdf?id=8GZxtu46Kx

[2] Murty et al. (2023). Grokking of Hierarchical Structure in Vanilla Transformers. https://aclanthology.org/2023.acl-short.38/

[3] Barak et al. (2022). Hidden Progress in Deep Learning: SGD Learns Parities Near the Computational Limit. https://openreview.net/pdf?id=8XWP2ewX-im

[4] Tirumala et al. (2022). Memorization Without Overfitting: Analyzing the Training Dynamics of Large Language Models. https://proceedings.neurips.cc/paper_files/paper/2022/hash/fa0509f4dab6807e2cb465715bf2d249-Abstract-Conference.html

[5] Gromov (2023). Grokking modular arithmetic. https://arxiv.org/abs/2301.02679

---

> ### Author Rebuttal · Authors · 2024-05-31
>
> Thank you for the valuable comments that help us improve our work. Here is our response:
>
> > Could engage with the relevant literature much more thoroughly.
>
> We appreciate you pointing out the additional related work that we missed. We will incorporate these references into the appropriate section in our revised version.
>
> > Writing of Section 5
>
> In Section 5, we did not intend to imply that implicit multi-task learning during the pretraining stage is the sole reason for the emergence of abilities. We agree that the emergence of abilities in current LLMs is driven by a multitude of complex factors and requires further investigation. Our goal with the simulation experiment was to offer an alternative perspective on understanding emergent abilities in LLMs, potentially spurring further research. We acknowledge that some of our wording may have been too strong, leading to misunderstandings, and will clarify this in the revised version.
>
> > Shape of memorization capacity
>
> Figure 3 shows a non-linear relationship between the model's hidden size and memorization capacity. Initially, memorization capacity increases rapidly with small hidden sizes, then transitions to a linear increase. The curve's flattening towards the end might be misleading due to the limited training data (12,769), which underestimates the model's capacity. Therefore, a definitive conclusion requires further verification with different training settings, such as various learning rates, data quantities, and task types. Further research on this topic would be interesting.
>
> Regarding the impact of layer number on memorization capacity, we conducted experiments with a hidden size of 32. The results are shown below. We observed that increasing the number of layers does increase the model's memorization capacity, but this increase is less significant compared to increasing the hidden size. For instance, comparing row 2 and row 5, despite row 2 having more parameters, it exhibits lower memorization capacity.
>
> | Hidden Size | Layer Num | Param Num | Memorization |
> | -- | -- | -- | -- |
> | 32 | 1 |  19,776 | 4,439 |
> | 32 | 2 | 32,064 | 6,183 |
> | 32 | 4 | 56,640 | 9,448 |
> | 40 | 1 | 28,560 | 6,697 |
> | 56 | 1 | 50,736 | 1,1341 |
>
> > Legend in Figure 4;  Use fewer evaluative words/phrases; Typos
>
> Thank you for highlighting these issues. We will make the necessary modifications to address all these problems.

---

> > ### Comment · Reviewer_q1HD · 2024-06-04
> > **Thank you**
> >
> > Thanks for the response. It sounds like some of the more straightforward writing concerns (R1 and R2) will be addressed. I also appreciate the inclusion of these new results; these make Fig. 3 even more interesting! That said, R3 still holds.
> >
> > Overall, I believe that this paper makes an interesting contribution and will generate interesting discussion at the conference if accepted. Given this and the responses to other reviewers, I am raising my score to an 8.

---

### Official Review · Reviewer_LPg7 · 2024-05-21

**Rating:** 6
**Confidence:** 3
**Ethics Flag:** 1

**Summary:**

This paper conducts an investigation into algorithmic tasks with a focus on the balance between memorization and generalization within deep neural networks. With a proposed theoretical framework grounded in two main hypotheses concerning the dataset size's criticality and the model's memorization capacity, the authors offer a graphical depiction of four training dynamics: progression, memorization, semi-grokking, and grokking. This framework is employed to illustraet the double descent phenomenon and link it with grokking, while also exploring multi-task learning, highlighting how generalization capabilities in algorithm tasks are significantly enhanced as model size increases.

**Questions To Authors:**

see weaknesses.

**Reasons To Accept:**

The paper provides theoretical view that provides a fresh lens for examining model performance across different training dynamics, particularly illuminating the elusive double descent phenomenon and its association with grokking. Additionally, the extension of analysis into the multi-task learning paradigm offers critical insights into the implications of model size for emergent abilities in large language models—demonstrating considerable depth and novelty in its approach to understanding deep neural models.

**Reasons To Reject:**

Major one: The empirical substantiation of the framework's applicability across varied model architectures and dataset complexities, as the current examination might be perceived as somewhat constrained, and its real-world applicability and practical implications remain a bit unclear. Empirical validation, especially in more realistic situations and model families (e.g., LLMs), could provide a more robust underpinning for the theoretical claims made. Can you try to empirically verify this framework on small-size language models such as GPT-2 (124M) or small-size diffusion models?

---

> ### Author Rebuttal · Authors · 2024-05-31
>
> Thank you for the valuable comments that help us improve our work. Here is our response:
>
> We acknowledge that a limitation of our study is that all experiments and analyses are conducted on algorithmic tasks, as noted in our *Limitations Section*. The primary reason for choosing these tasks is their simplicity, absence of noisy data, and lack of overlap between the training and validation sets. This allows us to clearly distinguish the model’s behaviors in terms of memorization and generalization.
>
> Moreover, we believe our framework can also make sense for more realistic models and tasks, as the concepts of memorization and generalization remain pertinent in these contexts. Notably, there is existing research that has observed grokking in real-world tasks such as MNIST and IMDB, though the effects are less pronounced compared to algorithmic tasks [1]. We are currently attempting to conduct experiments on real-world tasks to further validate our framework. However, these experiments are highly time-consuming due to the extremely long training periods required for observing grokking. Should we obtain additional results, we will be pleased to include them in our revised paper.
>
> [1] Ziming Liu, Eric J. Michaud, Max Tegmark. Omnigrok: Grokking Beyond Algorithmic Data. ICLR 2023

---

### Decision · Program_Chairs · 2024-07-10

**Decision:**

Accept

**Comment:**

The reviewers are largely positive, seeing the theoretical construction and the empirical results as highly valuable. The paper is interesting and should improve our understanding of several well-studied phenomena.

The paper would be improved by weakening its claims to account for the possibility of other factors in abrupt emergence. In particular, I find the consistency of a "memorized" vs "generalized" circuit to be overstated; there are many heuristics that can be used for generalization, which are not necessarily described as memorization, as seen in the recency/positional vs hierarchical heuristics in [1]. The empirical findings, other than the added scale factor, are not novel, but the framing of those findings is.

[1] Kabir Ahuja, Vidhisha Balachandran, Madhur Panwar, Tianxing He, Noah A. Smith, Navin Goyal, Yulia Tsvetkov (2024). Learning Syntax Without Planting Trees: Understanding When and Why Transformers Generalize Hierarchically. https://arxiv.org/abs/2404.16367

[comment from PCs] Please carefully consider the AC comments and follow their suggestions.